# Women's education and coverage of skilled birth attendance: An assessment of Sustainable Development Goal 3.1 in the South and Southeast Asian Region

**Jahar Bhowmik[1], Raaj Kishore Biswas[2]\*, Nurjahan Ananna[3]**

**1** Department of Health Science and Biostatistics, Swinburne University of Technology, Melbourne, VIC, Australia, **2** Transport and Road Safety (TARS) Research Centre, School of Aviation, University of New South Wales, Sydney, Australia, **3** Ibrahim Medical College, Dhaka, Bangladesh

⊜ These authors contributed equally to this work.
\* RaajKishore.Biswas@student.unsw.edu.au

**Data Availability Statement:** The secondary data sets analyzed in the current study are freely available upon request from the DHS website (https://dhsprogram.com/data/available-datasets.

## Abstract

### Objective

The objective of Sustainable Development Goal 3.1 is to reduce the global maternal mortality ratio (MMR) below 70 per 100,000 live births by 2030. One of the indicators for this objective is the proportion of births attended by skilled health attendants (SBA). This study assessed the progress of low- and middle-income countries from South and Southeast Asian (SSEA) region in SBA coverage and evaluated the contribution of women's education in this progression.

### Methods

The Demographic and Health Surveys were assessed, which included 38 nationally representative surveys on women aged between 15-49 years from 10 selected SSEA region countries in past 30 years. Binary Logistic regression models were fitted adjusting the survey clusters, strata and sampling weights. Meta-analyses were conducted by collapsing effect sizes and confidence intervals of education modeled on SBA coverage.

### Results

Results indicated that Cambodia, Indonesia and Philippines had over 80% SBA coverage after 2010, whereas Bangladesh and Afghanistan had around 50% coverage. Women with primary, secondary and higher level of education were 1.65, 2.21 and 3.14 times significantly more likely to access SBA care during childbirth respectively as compared to women with no education, suggesting that education is a key factor to address skilled delivery cares in the SSEA region.

cfm). Permission for this project was taken from the Demographic and Health Surveys (DHS) Program authority by the authors.

**Funding:** The authors received no specific funding for this work.

**Competing interests:** The authors have declared that no competing interests exist.

## Conclusion

Evaluation of the existing skilled birth attendance policies at the national level could provide useful insight for the decision makers to improve access to skilled care at birth by investing on women's education in remote and rural areas.

## Introduction

The Sustainable Development Goals (SDGs) are a collection of 17 global goals set by the United Nations (UN) General Assembly in 2015 for the year 2030. Progress in the public health domain in low- and middle-income countries (LMICs) are typically assessed using the targets of SDGs. The aim of SDG 3 is to *ensure healthy lives and promote well-being for all at all ages.* The first sub-goal of SDG 3 is goal 3.1: "reduce the global maternal mortality ratio to less than 70 per 100,000 live births" by 2030. One of the two indicators for this objective is the proportion of births attended by skilled health personnel and the other being maternal mortality ratio (MMR) [1].

Although the maternal mortality ratio, the proportion of mothers that do not survive childbirth compared to those who do, declined by 37 percent between 2000 and 2015, there were approximately 303,000 maternal deaths worldwide in 2015, most were due to preventable causes [2]. Compared to the developed regions, maternal mortality ratio is still 14 times higher in developing regions, which indicates that greater focus is required for the developing regions to achieve the SDGs by 2030 [3].

A limitation in assessing SDG progress in the SSEA countries is data paucity or irregular national level data [4]. According to this UN report, the Asia-Pacific region has made "satisfactory progress on" SDG 3. This includes the proportion of births attended by skilled health personnel, and this progress needs to be maintained to meet the 2030 target [4, 5]. However, performance of the developing countries varied across goals, countries, and regions in attaining Millennium Development Goals (MDG), which is expected to continue for SDGs as well [6]. Furthermore, consorted collaborative efforts from both government and experts are required to monitor the progress and to assess implementation strategies [7], which demands evaluation of national data sets on the progress of SDGs.

SBA is a broad category that encompasses health professionals, particularly doctors, nurses, and midwives, who are certified to attend mothers and newborn babies prior to and during delivery to manage normal deliveries and diagnose, manage or refer obstetric complications [8–11]. To keep consistency with the definitions used in the Demographic and Health Surveys (DHS), this study considered a qualified doctor, nurse, midwife, paramedic, family welfare visitor (FWV), community skilled birth attendant (CSBA), or sub-assistant community medical officer (SACMO) as SBA [12]. Orthodox village doctors without academic qualifications, uncertified community workers, and untrained conventional midwives were identified as traditional birth attendants (TBAs).

There has been an increasing number of studies that investigated factors associated with health-seeking behaviors of mothers and children [13]. Similar findings across these studies suggested that maternal health care is generally affected by various personal, sociocultural and environmental factors, including individual perceptions of health, self-efficacy, motivation, social values and belief systems [14–16]. Access to SBA services are found to be associated with various sociodemographic factors, for example education of women/mothers, religion, residency (urban/rural) and household financial capabilities are important predictors for women's access to maternal health services [17–20].

One important contributor to public health success in LMICs is education of women [21, 22]. Past two decades have observed growth in education, both for men and women in SSEA countries with high socioeconomic return [23, 24]. Multiple governmental and non-governmental programs were conducted using both foreign aid and public funding to increase school enrollments, with particular focus on women's education [25, 26]. One objective of these programs, and overall literacy rate, is to inform women of their rights and literate them on maternal health care including access to SBA [27, 28].

Findings on past studies demonstrate that improving SBA coverage rate could significantly reduce maternal and child deaths, particularly adopting reinforcement of the programs focusing on training for health personnel and education for mothers [27, 29, 30]. Similarly, it is expected that education of mothers would contribute to the increased access to SBA as educated mothers are likely to be more cautious about the complications could occurred during delivery time and more likely to access modern health care [31–33]. However, there is a gap in literature in quantifying the associated sociodemographic factors that commonly influence SBA accessibility across the SSEA region, particularly focusing the objective of attaining SDG 3.1.

This study focuses on the skilled birth attendance of the selected developing countries in the SSEA region, for which representative data sets on the population level are available through the DHS program. These include surveys from Afghanistan, Bangladesh, Cambodia, India, Indonesia, Maldives, Myanmar, Nepal, Pakistan, Philippines, and Timor-Leste. The primary objective of this study is to assess the progress of SDG 3.1 using proportion of skilled birth attendance and to investigate the sociodemographic factors associated with the gradual increase of skilled birth attendants (SBA) across these countries. More specifically, effects of women's education in the progress of SDG 3.1 was evaluated through its association with SBA coverage within the selected SSEA countries.

Although several studies used DHS data sets from African nations to evaluate maternal health services, evaluation on the cross-country assessment in the SSEA region is limited [34–38]. For a consistent nationwide data collection process with similar methodologies followed in the selected LMICs, this study was limited to DHS surveys between 1990 and 2017. A meta-analysis was conducted to estimate the overall association between women's education on SBA for the selected countries in the SSEA region.

## Theoretical framework

Along with evaluating the overall SBA coverage in the selected countries in SSEA region, the primary hypotheses of the current study are; women's education is associated with access to SBA, and the SBA coverage will increase with mother's level of education. Several sociodemographic factors were included in the statistical models as covariates after evaluating past studies, where they were found to be significantly associated with SBA coverage. However, a theoretical road map was constructed to apprehend the complex associations between ecological factors and maternal health care.

This study followed the Person-Centered Care Framework for Reproductive Health Equity (PCRHC) [39], which is used for maternal health care research in LMICs [40]. There are three contexts for PCRHC in reproductive health care: a) societal and community determinants, b) women's health-seeking behaviors, and c) the quality of care in the facility [39]. The first two contexts of Person-Centered Maternity Care, we hypothesized, were associated with women's level of literacy, particularly in LMICs [41, 42].

Based on the existing literature, [39] listed eight domains of PCRHC: autonomy, confidentiality, communication, social support, supportive care, trust and health facility

environment. Women's education along with the selected sociodemographic factors were expected to be directly or indirectly associated with these domains [43, 44].

## Materials and methods

### Data overview

DHSs are considered standardized and nationally representative cross-sectional surveys, which has been conducted in LMICs since 1984 [45, 46]. As the survey methodology is consistent across DHS and collected variables are identical, these surveys allow assessments over multiple populations over the time. All DHSs follow a standard protocol with consent from the human participants approved by ICF Macro Institutional Review Board and local research ethics committee. The authors had access to de-identified survey data with permission from Measure DHS and ICF (approval number: 127313). The secondary data sets analyzed in the current study are freely available upon request from the DHS website at http://dhsprogram. com/data/available-datasets.com.

Every survey conducted by DHS followed a two-stage stratified cluster sampling technique [12, 47]. Sampling frame consists of a list of enumeration areas (EAs) using recent census data. For first stage, EAs (or clusters) are selected using probability proportional to size (PPS) sampling method, where the number of clusters/EAs vary across countries. For example, typically there are 600 clusters in Bangladesh and 28,522 clusters in India. An equal probability systematic sampling method is applied in the second stage to select a pre-specified number of households from each cluster. Generally, the survey focuses on women of reproductive health age group (15-49 years), although some surveys included men as well. In this current study, only data from female respondents were extracted from the selected surveys.

### Surveys

From the selected 10 SSEA countries, surveys from 1990 to 2017 were included in this study. Data from a total of 38 surveys containing 1,171,731 participants (women) were analyzed. The included surveys for the 10 countries are Afghanistan (2015), Bangladesh (1993, 1996, 1999, 2004, 2007, 2011, 2014), Cambodia (2000, 2005, 2010, 2014), India (1992, 1998, 2006, 2015), Indonesia (1997, 2002, 2007, 2012), Myanmar (2016), Nepal (1996, 2001, 2006, 2011, 2016), Pakistan (1990, 2006, 2012, 2017), Philippines (1993, 1998, 2003, 2008, 2013, 2017), Timor Leste (2009, 2016). The surveys those were not typical DHS (e.g., Afghanistan mortality survey 2010, Cambodia special DHS 1998) or contained incomplete data (Indonesia 1991 or 1994) were excluded from the analysis due to lack of necessary variables. Data from Maldives were also excluded as they had 95% and 100% SBA coverage in 2009 and 2016 surveys respectively.

### Variable

In this study, access to SBA is considered as the outcome variable. DHS VI standard recode manual were followed while defining SBA [45]. It was recoded as a binary variable with women who had accessed SBA vs those who did not. As explained earlier, qualified doctor, nurse, midwife, paramedic, FWV, MA and SACMO were considered as skilled ANC providers and SBAs. For one respondent who seek multiple services, the one with the highest qualification was considered as birth attendant during delivery.

According to past literature and outcome from the pre-analysis results (missing values and consistency of variables in the surveys over the years), seven sociodemographic factors were included in this study as explanatory variables [48–50]. The selected explanatory variables are age of respondents (continuous measured in in years); residence (urban, rural); education of

both respondent and her partner (No education, primary, secondary, higher); wealth index (poorest, poorest, middle, richer, richest); age at first birth (years); and age of partner/husband (years). For model adjustment purpose, survey weights, strata and cluster information were also extracted.

Education is defined by whether the respondent attended school and if so, the number of years of schooling [51]. Based on this information, DHS provides a standardized variable with four categories mentioned above adjusting for the country-wise education system [52]. For example, the threshold for primary and secondary education in Bangladesh is grade 5 and 10 [53], whereas they are grade 6 and 12 in Afghanistan [54]. The standardized categorization coding is used so that comparison across surveys are possible [55]. Similarly, the wealth index is a standardized measure, quantified using principal component analysis (PCA) from household assets [45].

## Statistical analysis

There are multiple approaches of combining the surveys and quantifying the associations between education and SBA access such as collapsing the surveys to a single data set and apply relevant regression approach or separately analyze each survey data set and merge the effect sizes with meta-analysis techniques. The latter was applied in this study primarily because country-wise (between subject) variation could not be adjusted even with multiple random effects in the models as countries are inherently different with heterogenous size, population, and some unobserved household characteristics. Also, sample size variation could bias the outcomes, for example, sample size in India alone is greater than all other combined surveys. Furthermore, number of surveys varied among countries which might lead to over or underrepresentation of some countries over the rest. Thus, to keep the survey integrity, each survey was modelled individually and then combined later through meta-analysis.

As the outcome variable was binary, a regression model with binomial family of distributions was adopted to find the association between SBA and sociodemographic factors. As DHS data were collected at multiple levels (cluster, strata and individual), generalized linear models (GLMs) with binary outcome were used in this study adjusting for cluster-wise and strata-wise effects. For generalization of the results, survey weights for each individual were adjusted in the GLMs, commonly used in DHS surveys [33]. As large-scale surveys were used in the study, missing cases were deleted list-wise with the assumption that data were missing at random for unbiased estimates [56]. The models were fitted using *R*-package *svyglm*(*survey*) [57].

Using the adjusted odds ratio of women's education status (primary, secondary and higher compared to no education) a meta-analysis was conducted for all surveys that indicated the association between education and SBA coverage. *R*-package *metafor* was used for fitting fixed effect forest plots. All data compilations and analyses were conducted in *R* (3.5.0) [58].

## Results

All 10 countries selected in this study have increased the SBA coverage over the years (Fig 1); however, annual improvement for the Afghanistan and Myanmar could not be observed as there were only one survey conducted during the selected period (1990—2017). The highest coverage was noted in Cambodia in 2014 survey (89%), apart from Maldives where 100% coverage was observed but not added in this study. India, Indonesia and Philippines also had over 80% SBA coverage according to 2012 survey. Only Bangladesh had below 50% SBA coverage (44.7% in 2014) among the selected countries in their latest surveys. The results were crosschecked with the individual survey's report.

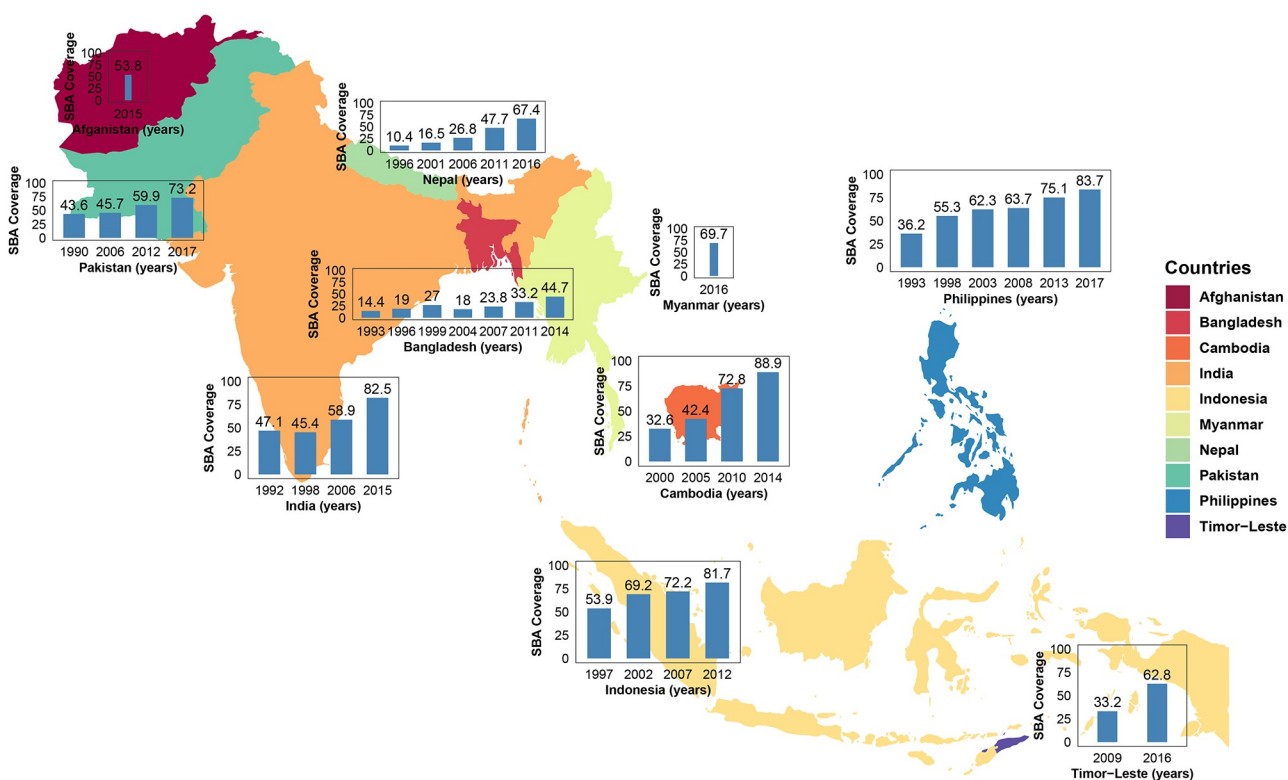

**Fig 1. The coverage of skilled birth attendants (in %) for 10 selected countries in the SSEA region during the period 1990-2017.**

Among the 10 selected countries, respondents from Afghanistan and Bangladesh had a small proportion of highly educated women with 85% of the surveyed women in Afghanistan in 2015 were uneducated (Table 1). Highest proportion of women belonged to secondary level of education in the recent surveys of India, Indonesia, Nepal, Philippines and Timor Leste. Over 50% of respondents in Pakistan were uneducated and women with primary level of education comprised 49% and 44% of total study sample in Cambodia and Myanmar respectively.

Most of the sociodemographic factors showed significant associations with access to SBA services. As the primary focus of this study is to evaluate the association between education of participants (women) and access to SBA services, the effect sizes of education on SBA coverage were extracted from each survey through a GLM outcomes. The association between SBA (dichotomous) coverage and education of participants categories are displayed in Figs 2–4 through forest plots. Adjusted odds ratio (AOR) of the women with primary, secondary and higher education compared to women with no education showed that most of the women with primary and secondary levels of education and all women with higher-level education status were significantly associated with their access to SBA services.

It is evident that the AOR increased with increased level of education (Figs 3 and 4). Summary estimates indicate that women with primary, secondary and higher level of education were 1.65, 2.21 and 3.14 times significantly more likely to access SBA during childbirth respectively. However, a greater variation in the effect sizes for the estimates of higher education were observed, as some surveys estimated increased impact of higher education on access to SBA services (e.g., Afghanistan 2015 and Indonesia 2007). It is to be noted that due to the small sample size of higher education category, inflated AOR and CI were detected for Cambodia (2000, 2005, 2010, 2014), Pakistan (1990), and Nepal (2006) surveys, and these were not

**Table 1. Distribution of women's level of education over 30 surveys across 10 selected countries the SSEA region.**

| Country | Level of Education | Survey years | | | | | | |
|---|---|---|---|---|---|---|---|---|
| | | 1990-1995 | 1996-1998 | 1999-2002 | 2003-2006 | 2007-2010 | 2011-2013 | 2014-2017 |
| | | Sample size (%) | | | | | | |
| Afghanistan | No education | - | - | - | - | - | - | 16818 (85.1) |
| | Primary | - | - | - | - | - | - | 1404 (7.1) |
| | Secondary | - | - | - | - | - | - | 1249 (6.3) |
| | Higher | - | - | - | - | - | - | 303 (1.5) |
| Bangladesh | No education | 1983 (55.5) | 2500 (54.3) | 2261 (43.6) | 1866 (34.8) | 1268 (25.8) | 1332 (18.2) | 607 (13.5) |
| | Primary | 1019 (28.5) | 1295 (28.1) | 1508 (29.1) | 1648 (30.7) | 1506 (30.6) | 2193 (29.9) | 1235 (27.5) |
| | Secondary | 497 (13.9) | 685 (14.9) | 1160 (22.4) | 1512 (28.2) | 1742 (35.4) | 3174 (43.3) | 2130 (47.4) |
| | Higher | 75 (2.1) | 123 (2.7) | 260 (5.0) | 339 (6.3) | 406 (8.2) | 626 (8.5) | 522 (11.6) |
| Cambodia | No education | - | - | 2205 (36.4) | 1700 (27.7) | 1314 (20.4) | - | 803 (13.6) |
| | Primary | - | - | 3102 (51.2) | 3495 (56.9) | 3381 (52.5) | - | 2914 (49.4) |
| | Secondary | - | - | 743 (12.3) | 920 (15.0) | 1628 (25.3) | - | 1968 (33.4) |
| | Higher | - | - | 8 (0.1) | 27 (0.4) | 121 (1.9) | - | 215 (3.6) |
| India | No education | 21909 (58.9) | 14706 (50.8) | - | 14082 (38.2) | - | - | 55104 (28.9) |
| | Primary | 5885 (15.8) | 4604 (15.9) | - | 5249 (14.3) | - | - | 26692 (14.0) |
| | Secondary | 7908 (21.3) | 6992 (24.2) | - | 14207 (38.6) | - | - | 88826 (46.6) |
| | Higher | 1481 (4.0) | 2622 (9.1) | - | 3286 (8.9) | - | - | 20145 (10.6) |
| Indonesia | No education | - | 1401 (10.2) | 604 (4.5) | - | 611 (4.0) | 423 (2.8) | - |
| | Primary | - | 6932 (50.5) | 6029 (45.3) | - | 6007 (39.2) | 4689 (30.8) | - |
| | Secondary | - | 4852 (35.3) | 5800 (43.6) | - | 7499 (48.9) | 8134 (53.5) | - |
| | Higher | - | 546 (4.0) | 868 (6.5) | - | 1203 (7.9) | 1966 (12.9) | - |
| Myanmar | No education | - | - | - | - | - | - | 622 (16.1) |
| | Primary | - | - | - | - | - | - | 1698 (43.9) |
| | Secondary | - | - | - | - | - | - | 1245 (32.2) |
| | Higher | - | - | - | - | - | - | 302 (7.8) |

(*Continued*)

**Table 1.** (Continued)

| Country | Level of Education | Survey years | | | | | | |
|---|---|---|---|---|---|---|---|---|
| | | 1990-1995 | 1996-1998 | 1999-2002 | 2003-2006 | 2007-2010 | 2011-2013 | 2014-2017 |
| | | Sample size (%) | | | | | | |
| Nepal | No education | - | 2645 (79.3) | 3021 (72.8) | 2455 (58.7) | - | 1765 (43.3) | 1231 (30.7) |
| | Primary | - | 380 (11.4) | 585 (14.1) | 745 (17.8) | - | 817 (20.0) | 763 (19.0) |
| | Secondary | - | 271 (8.1) | 497 (12.0) | 856 (20.5) | - | 1225 (30.0) | 1396 (34.8) |
| | Higher | - | 41 (1.2) | 45 (1.1) | 126 (3) | - | 272 (6.7) | 616 (15.4) |
| Pakistan | No education | 3066 (76.6) | - | - | 3802 (66.6) | | 4112 (55.3) | 4178 (50.4) |
| | Primary | 370 (9.2) | - | - | 782 (13.7) | - | 1062 (14.3) | 1101 (13.3) |
| | Secondary | 500 (12.5) | - | - | 762 (13.4) | - | 1368 (18.4) | 1747 (21.1) |
| | Higher | 65 (1.6) | - | - | 361 (6.3) | - | 899 (12.1) | 1261 (15.2) |
| Philippines | No education | 382 (2.5) | 174 (3.3) | - | 99 (2.0) | 88 (1.9) | 101 (1.9) | 117 (1.5) |
| | Primary | 4863 (32.4) | 1825 (34.9) | - | 1422 (28.9) | 1142 (24.8) | 1120 (21.2) | 1521 (19.0) |
| | Secondary | 5868 (39.1) | 1941 (37.1) | - | 2052 (41.8) | 2155 (46.7) | 2585 (48.8) | 3945 (49.4) |
| | Higher | 3913 (26.0) | 1296 (24.8) | - | 13401 (27.3) | 1227 (26.6) | 1486 (28.1) | 2409 (30.1) |
| Timor Leste | No education | - | - | - | - | 2022 (33.7) | - | 1196 (24.3) |
| | Primary | - | - | - | - | 1716 (28.6) | - | 912 (18.6) |
| | Secondary | - | - | - | - | 2157 (36.0) | - | 2377 (48.4) |
| | Higher | - | - | - | - | 104 (1.7) | - | 431 (8.8) |

added in forest plots (Fig 4). For assessing multicollinearity, the variance inflation factor (VIF) scores were quantified for each survey and most of the VIFs scores from the models were under 5 which suggests no multicollinearity [59]. However, VIF was over 5 (but below 10) for partner's level of education in the surveys of Indonesia and education of respondent and partners in surveys of Philippines, which would not pose major multicollinearity issues [60].

## Discussion

The aim of this study was to estimate SBA coverage in the 10 selected countries from the SSEA region during the period 1992-2017 and to evaluate the association of women's education with the SBA coverage, which ultimately link with SDG 3.1. Results obtained by analyzing 38 DHS data sets showed that all 10 selected countries in the SSEA region have improved their access to SBA coverage during the surveys period. However, the trend of improvement on SBA coverage was not homogeneous across the region; as over 80% SBA coverage was observed in Cambodia, India, Indonesia and Philippines in the latest surveys. However, Bangladesh (44.7%) and Afghanistan (53.8%) only had around 50% nationwide SBA coverage. Further analysis

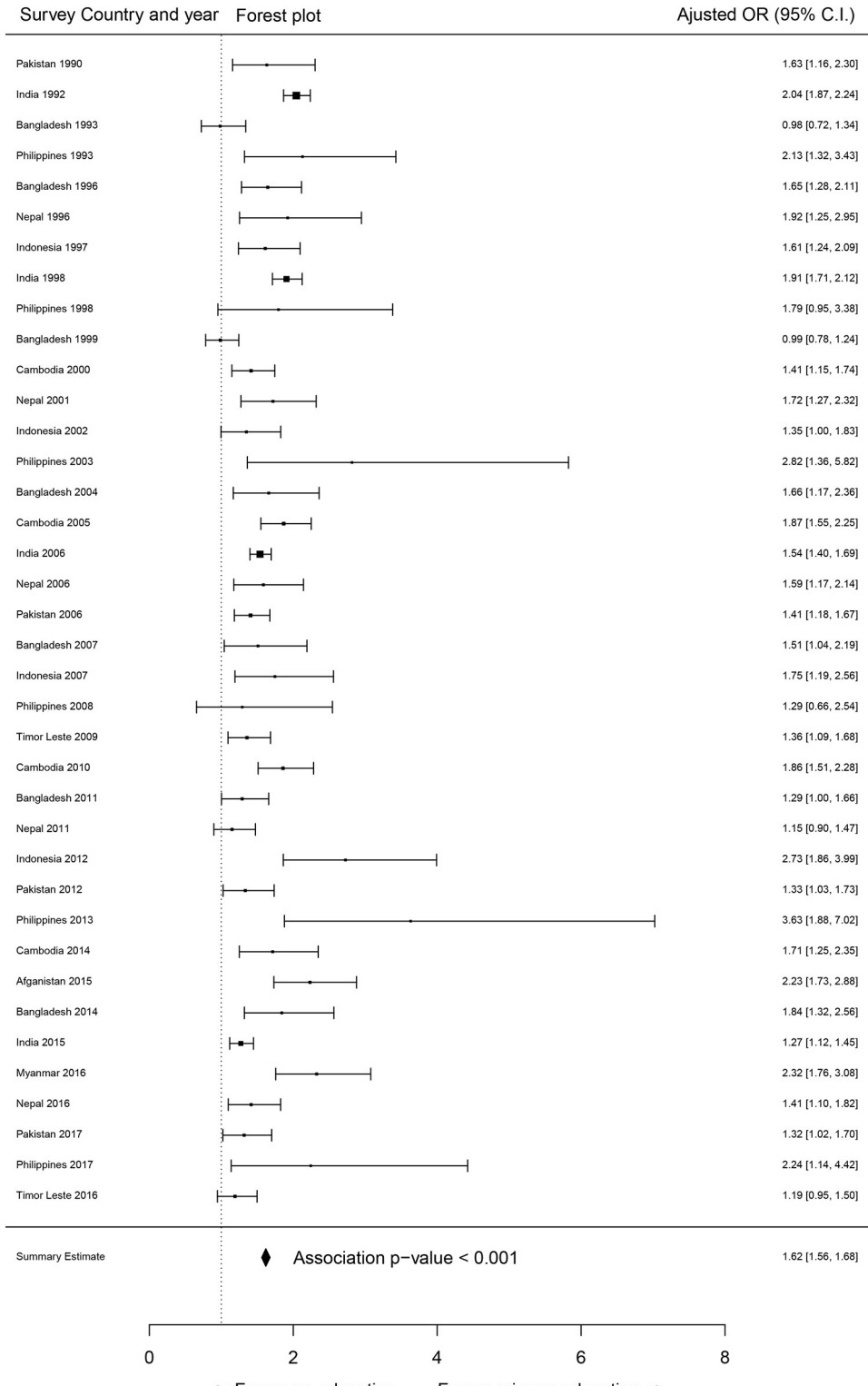

**Fig 2. Forest plot of odds of women having primary education on access to skilled birth attendants compared to women with no education (OR = odds ratio, C.I. = confidence interval).**

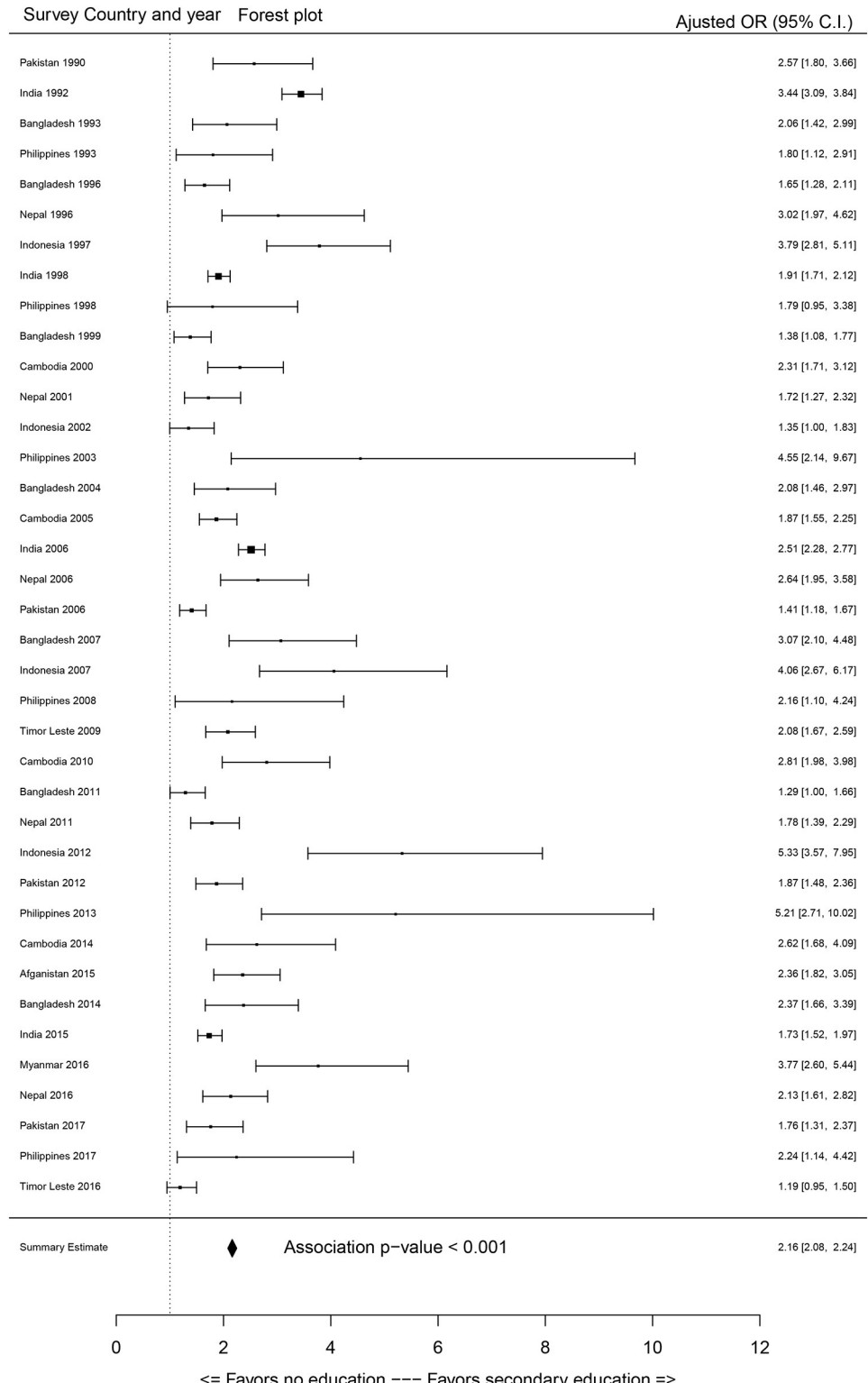

**Fig 3. Forest plot odds of women having secondary education on access to skilled birth attendants compared to women with no education (OR = odds ratio, C.I. = confidence interval).**

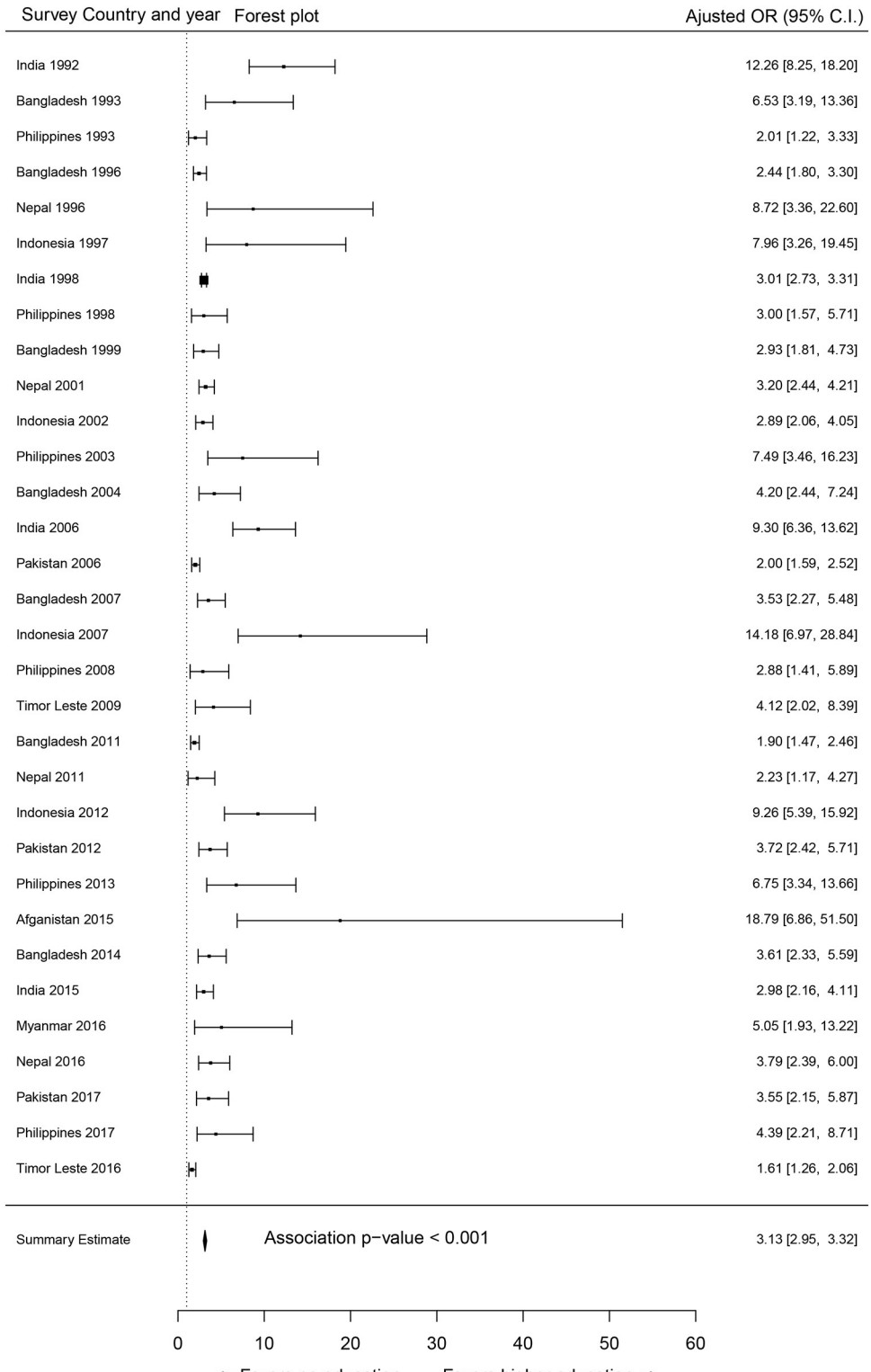

**Fig 4. Forest plot of odds of women having higher education on access to skilled birth attendants compared to women with no education (OR = odds ratio, C.I. = confidence interval).**

revealed that levels of education (primary, secondary and higher) for women were significantly associated with SBA coverage, where increased level of education lead to a better access to SBA services.

Autonomy of women plays a key role in person-centered health care. South and Southeast Asian families are typically male donated and oldest male is generally the house head. In patriarchal society, voice of women is limited to the privileged section of the society, generally higher educated women who contribute to the family economy [61, 62]. This traditional scenario and cultural norms bars women from going beyond typical health care (TBA in this case) and avail any modern health care on their own [63, 64]. However, it is argued that educated women are likely to make their own health care decisions and firmer to reject the ancestral delivery methods [65–67]. Thus, allowing access to education seem to help women partake SBA services and ultimately contribute to lowering the risk of maternal deaths.

It is well-known that health care awareness is commonly associated with level of formal education. This is supported by the findings of this current study. Results showed that mothers with greater access to education were more likely to seek SBA services compared to their illiterate counterparts. Women, who completed their secondary and higher level of education, are expected to be well-informed about the various health issues, particularly the problems of seeking traditional unscientific cheap treatments [68–70].

An educated woman is more likely to understand the consequences of traditional medicines or services from TBA, usually in rural areas that lead to unattended home deliveries [18, 71]. To rise above the long-established mindset of the community and rebel against the conventional flow to seek modern medical help, higher education would be necessary to avert maternal deaths [72].

In LMICs, education is typically correlated with family's economic status. Educated families are more likely to be financially stable. Due to the cost involved, SBA services or most other maternal health care services are often limited to well-off families where women are educated [73, 74]. Hospital admissions or access to any means of health care are considered a financial burden for insolvent households and compel them to use services from TBAs, who provide cheaper services in the locality [18, 75–77].

The scenario worsens with limited number of health care professionals work in remote and rural areas, which is particularly applicable for countries such as Afghanistan, Bangladesh, India and Pakistan [78–80]. These add to the extra travel cost to access SBA services, which again is cheaper through home delivery by a TBA [81, 82]. These were also supported by the findings obtained in the current study, as women with highest level of education were three times more likely to access modern delivery cares compared to women with no education, which also indicated that primary or basic level of education might not be enough for achieving the target of SDG 3.1.

Although this study analyzed data with a large sample size, there were few limitations need to be noted. Firstly, all DHS data are cross-sectional, which limits the scope of casual interpretation of the findings. Future studies could consider conducting a counterfactual analysis to determine lives saved or maternal deaths avoided due to the increase in the proportion of educated women in the SSEA region. Secondly, calculation of SBA in surveys varied in terms of SBA coverage measured in years preceding the survey. For example, Bangladesh DHS 2014 used all deliveries 3 years preceding the survey, Afghanistan 2015 survey used 5 years preceding the survey and India 1992 survey used 4 years preceding the survey. Thirdly, during data cleaning, some strata in various surveys had complete missing or low sample count, which was omitted for model building purpose. Finally, as the data sets were interrupted time series, no prediction could be undertaken. However, future studies could apply interrupted time series models to compliment the meta-analysis and could consider forecasting. In addition, there

were other potential sociodemographic factors such as household wealth index and women's area of residence, as reported in the supplementary file (please refer to Tables 1-38), which render further discussion regarding SBA access but were not within the scope of this paper.

## Conclusions

With just over 10 years to go for 2030, deadline for achieving the SDGs, most LMICs are still struggling to keep up with the standards set. These countries must inject a sense of urgency by changing policies and executing accelerated actions at the national level. The study found heterogeneous trend in SBA coverage improvement in the SSEA region over the years with some countries already gained above 80% coverage and some remained below the halfway mark. Women's education, as hypothesized, was found to be significantly associated with access to SBA services in all 10 selected countries over 38 national surveys, which shows a reasonable link between literacy rate of women and access to maternal health services.

Multiple intervention programs focusing the vulnerable cohorts, especially uneducated mothers in regional areas (e.g. Afghanistan and Bangladesh) could help improving the coverage of skilled delivery care. Both public and non-governmental organizations could intensify SBA training programs and use awareness campaigns to increase the SBA coverage. Achieving the SDG 3.1 requires the partnership of governments, private sector, civil society and citizens alike to make sure women's education and their rights are prioritized, especially in the LMICs. Evaluation of the existing skilled birth attendance policies at the national level might provide useful insight for the decision makers to improve access to skilled care at birth by increasing women's education in remote and rural areas. Evidences from country-wise research using population data and analyzing regional trends could be applied to design better facilities including health infrastructure and accessibility that would contribute to achieving SDG 3.1.

## Note

The country names generally were inscribed in alphabetical order where they were tabulated or discussed as a list.

## Supporting information

**S1 Checklist.**
(DOCX)

**S1 File.**
(DOCX)

**S2 File.**
(DOCX)

## Acknowledgments

The authors would like to acknowledge MEASURE Evaluation Dataverse, which allows researchers to access their data for free.

Views expressed in this study do not necessarily reflect those of USAID, the US government, or MEASURE Evaluation.

## Author Contributions

**Conceptualization:** Jahar Bhowmik.

**Data curation:** Raaj Kishore Biswas.

**Formal analysis:** Raaj Kishore Biswas.

**Investigation:** Jahar Bhowmik, Raaj Kishore Biswas, Nurjahan Ananna.

**Methodology:** Raaj Kishore Biswas.

**Project administration:** Jahar Bhowmik.

**Resources:** Nurjahan Ananna.

**Software:** Raaj Kishore Biswas.

**Supervision:** Jahar Bhowmik.

**Validation:** Jahar Bhowmik.

**Visualization:** Raaj Kishore Biswas.

**Writing – original draft:** Raaj Kishore Biswas.

**Writing – review & editing:** Jahar Bhowmik, Nurjahan Ananna.

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
