## [Decision Letter · Decision Letter 0]

9 Jan 2020

PONE-D-19-30198

Women's education and coverage of skilled birth attendance: An assessment of Sustainable Development Goal 3.1 in the South and Southeast Asian Region

PLOS ONE

Dear Mr Biswas,

Thank you for submitting your manuscript to PLOS ONE. After careful consideration, we feel that it has merit but does not fully meet PLOS ONE’s publication criteria as it currently stands. Therefore, we invite you to submit a revised version of the manuscript that addresses the points raised during the review process.

We would appreciate receiving your revised manuscript by Feb 23 2020 11:59PM. To enhance the reproducibility of your results, we recommend that if applicable you deposit your laboratory protocols in protocols.io, where a protocol can be assigned its own identifier (DOI) such that it can be cited independently in the future. For instructions see: http://journals.plos.org/plosone/s/submission-guidelines#loc-laboratory-protocols

We look forward to receiving your revised manuscript.

Kind regards,

Calistus Wilunda, DrPH

Academic Editor

PLOS ONE

Additional Editor Comments (if provided):

Definition of education. The authors need to clarify how education was defined. The assumption is that the countries included in the study had a similar education system at the time of the surveys, with the same number of years of schooling at each level? If this is not true, then years of schooling may be a more consistent  variable than education level.

What was the extend of missing data (for all the variables included in the analyses)? Could this have affected the observed results? Was the pattern of “missingness” checked? Did you consider using statistical methods to handle missing data?

Please include a STROBE checklist as supplementary file.

Journal Requirements:

3. We note that Figure 1 in your submission contain map images which may be copyrighted. All PLOS content is published under the Creative Commons Attribution License (CC BY 4.0), which means that the manuscript, images, and Supporting Information files will be freely available online, and any third party is permitted to access, download, copy, distribute, and use these materials in any way, even commercially, with proper attribution. For these reasons, we cannot publish previously copyrighted maps or satellite images created using proprietary data, such as Google software (Google Maps, Street View, and Earth). For more information, see our copyright guidelines: http://journals.plos.org/plosone/s/licenses-and-copyright.

You may seek permission from the original copyright holder of Figure 1 to publish the content specifically under the CC BY 4.0 license. 

If you are unable to obtain permission from the original copyright holder to publish these figures under the CC BY 4.0 license or if the copyright holder’s requirements are incompatible with the CC BY 4.0 license, please either i) remove the figure or ii) supply a replacement figure that complies with the CC BY 4.0 license. Please check copyright information on all replacement figures and update the figure caption with source information. If applicable, please specify in the figure caption text when a figure is similar but not identical to the original image and is therefore for illustrative purposes only.

Reviewers' comments:

Reviewer's Responses to Questions

**Comments to the Author**

1. Is the manuscript technically sound, and do the data support the conclusions?

Reviewer #1: Yes

Reviewer #2: Partly

2. Has the statistical analysis been performed appropriately and rigorously? 

Reviewer #1: Yes

Reviewer #2: Yes

3. Have the authors made all data underlying the findings in their manuscript fully available?

Reviewer #1: Yes

Reviewer #2: Yes

4. Is the manuscript presented in an intelligible fashion and written in standard English?

Reviewer #1: Yes

Reviewer #2: Yes

5. Review Comments to the Author

Reviewer #1: General Comments

1. This is a well written manuscript. The methods and statistical analysis is well executed.

2. Check the font size of line 176 on page 7.

Supplementary material

3. The Tables in the supplementary material are very informative and I wish the authors will consider referring them or presenting some of them in the result section.

4. Why is the intercept reported in the table? I don’t think this is adding anything to the result hence consider removing it.

5. There is no need repeating the reference category for variable category.

6. I think the tables in the supplementary material can be presented in a better way. See example of Table 1 below.

7. Consider reporting all p-values to 3 decimal place.

Reviewer #2: Topic is highly relevant. Region which authors have selected is also quite relevant.

I have few comments on analysis done in the paper

2. Authors do not justify their selection of the method sufficiently.

3. There is a high likelihood that variables used as co-variates in the model are highly correlated. but Authors do not discuss this.

4. All 38 country-year data is used as separate regression. The idea behind having some many data-sets is to combine them in meaningful way in order to gain greater insights.

5. For Example, at the least, authors could create one single dataset after combining all surveys of given country conducted in different years to see how things have changed over time after introducing a time indicator. And this allow them to draw a comparison of countries in making progress over time that have more than one survey.

6. Since authors are using Multilevel method, they could have used variance of random components related to strata and clusters as additional sources of information to understand significance context in which individuals live, i.e., neighbourhood and regional factors may have effect on the choice of SBA by individual mothers.

6. PLOS authors have the option to publish the peer review history of their article (what does this mean?). If published, this will include your full peer review and any attached files.

Reviewer #1: No

Reviewer #2: Yes: Zakaria Siddiqui

---

## [Author Response · Author response to Decision Letter 0]

5 Feb 2020

5 February 2020

Calistus Wilunda

Academic Editor,

PLOS ONE

RE: PLOS ONE - Decision on Manuscript ID PONE-D-19-30198

Dear editor,

Thank you for proving us the opportunity to revise our manuscript entitled “Women's education and coverage of skilled birth attendance: An assessment of Sustainable Development Goal 3.1 in the South and Southeast Asian Region” for publication in PLOS ONE. 

We have addressed the points raised by the academic editor and reviewers in our responses, and incorporated related changes into the original manuscript through a comprehensive revision. For convenience, the reviewers’ comments are highlighted in blue and our responses are highlighted in black colours font. 

The primary changes made in the revised version of the manuscript include: 

1. Adding justification for using the chosen statistical approach.

2. Quantifying variance inflation factors for checking multicollinearity and reported as a supplementary file.

3. Addressing the definition of education and missing values.

4. Including STROBE checklist.

In addition to the above, all authors have read and revised the manuscript. Some minor changes throughout the revised version of the manuscript are made, mostly to improve clarity of expressions.

We would like to thank academic editor and reviewers for their valuable feedback and suggestions. 

Sincerely, 

Authors

Women's education and coverage of skilled birth attendance: An assessment of Sustainable Development Goal 3.1 in the South and Southeast Asian Region

(PONE-D-19-30198)

Reviewer #1

Reviewer’s comment

1. This is a well written manuscript. The methods and statistical analysis is well executed.

2. Check the font size of line 176 on page 7.

Authors’ response

Thank you for your encouraging comment. We have addressed the font size issue. 

Reviewer’s comment

3. The Tables in the supplementary material are very informative and I wish the authors will consider referring them or presenting some of them in the result section.

Authors’ response

Thanks for this valuable suggestion. The reason for not referring to the details of other results from the supplementary tables is primarily because these variables were not relevant to our research hypothesis. They were only used to adjust the model. Furthermore, commenting on them would require extending the analysis of these variables to combine their effect sizes using forest plots or a similar method. We suspect that it would distract the readers from the primary objective of the study. However, these could be avenues for future research to investigate this. Thus, we added the following sentence in the limitation section (line 283-287 at page 8 in the revised manuscript). 

“In addition, there were other potential sociodemographic factors such as household wealth index and women's area of residence, as reported in the supplementary file (please refer to Tables 1-38) , which render further discussion regarding SBA access but were not within the scope of this paper.”

Reviewer’s comment

4. Why is the intercept reported in the table? I don’t think this is adding anything to the result hence consider removing it.

5. There is no need repeating the reference category for variable category.

6. I think the tables in the supplementary material can be presented in a better way. See example of Table 1 below.

7. Consider reporting all p-values to 3 decimal place. 

Authors’ response

Thank you for your careful investigation and providing a detailed feedback. We have removed the results of intercept and amended the reference category by avoiding the repetition as per your example. We also ensured that p-values are consistent with 3 decimal places throughout the paper. Please refer to the attached revised supplementary document (Tables 1-38). 

Reviewer #2

Reviewer’s comment

Topic is highly relevant. Region which authors have selected is also quite relevant.

I have few comments on analysis done in the paper

2. Authors do not justify their selection of the method sufficiently.

Authors’ response

Thanks for this useful comment. We have addressed this in the ‘Statistical Analysis’ section, where we explained that GLM was used because the outcome variable was binary and also it is a common approach used for analyzing DHS data sets. However, we agree that we need to justify the approach used to model each survey separately through a meta-analysis. We have added the following paragraph in the revised version of the manuscript as justification (line 155-166 at page 5 of the revised manuscript).

“There are multiple approaches of combining the surveys and quantifying the associations between education and SBA access such as collapsing the surveys to a single data set and apply relevant regression approach or separately analyze each survey data set and merge the effect sizes with meta-analysis techniques. The latter was applied in this study primarily because country-wise (between subject) variation could not be adjusted even with multiple random effects in the models as countries are inherently different with heterogenous size, population, and some unobserved household characteristics. Also, sample size variation could bias the outcomes, for example, sample size in India alone is greater than all other combined surveys. Furthermore, number of surveys varied among countries which might lead to over or under-representation of some countries over the rest. Thus, to keep the survey integrity, each survey was modelled individually and then combined later through meta-analysis.” 

Reviewer’s comment

3. There is a high likelihood that variables used as co-variates in the model are highly correlated. but Authors do not discuss this.

Authors’ response

Thank you for raising this important issue. We assessed the variance inflation factor (VIF) scores and most were below 5, which indicated that multicollinearity was not an issue for most of the fitted models (Kutner, M.H., Nachtsheim, C. and Neter, J., 2004. Applied linear regression models. McGraw-Hill/Irwin). However, partner’s level of education in the surveys of Indonesia and education of respondent and partners in surveys of Philippines showed multicollinearity. Please see the attached VIF tables included in the supplementary file. We have addressed this on page …(line……) of the revised version of the manuscript.

“For assessing multicollinearity, the variance inflation factor (VIF) scores were quantified for each survey and most of the VIF scores obtained from the models were under 5 (conservative) which suggests no multicollinearity [57]. However, VIF was over 5 (but below 10) for partner’s level of education in the surveys of Indonesia and education of respondent and partners in surveys of Philippines, which would not pose major multicollinearity issues [60].”

Reviewer’s comment

4. All 38 country-year data is used as separate regression. The idea behind having some many data-sets is to combine them in meaningful way in order to gain greater insights.

5. For Example, at the least, authors could create one single dataset after combining all surveys of given country conducted in different years to see how things have changed over time after introducing a time indicator. And this allow them to draw a comparison of countries in making progress over time that have more than one survey.

Authors’ response

We have addressed this issue in the first response (Reviewer #2) It is primarily because a single dataset collapsing all 38 surveys might not be able to adjust country wise variations. A paragraph clarifying this inserted on page 5 (line 155-166) of the revised version of the manuscript. 

Reviewer’s comment

6. Since authors are using Multilevel method, they could have used variance of random components related to strata and clusters as additional sources of information to understand significance context in which individuals live, i.e., neighbourhood and regional factors may have effect on the choice of SBA by individual mothers.

Authors’ response

We have used generalized linear model (GLM) where cluster, strata and weights were adjusted for each survey. No random term in the regression models were included. This approach is used for complex survey design with multiple level of subpopulations (e.g., strata/cluster). For details please refer to: https://doi.org/10.18637/jss.v009.i08. The author, who also designed the ‘survey’ package which we used for modelling, noted “For stratified designs the JKn jackknife removes one PSU at a time, but reweights only the other PSUs in the same stratum.” So, population corrections are used while estimating the effect sizes. 

The neighborhood variation could be analyzed by collapsing all the surveys to one single dataset and then use country as a random effect. However, as explained above, those variations would not be explainable given the size, population and household level heterogeneities exist among the countries. 

Additional Editor Comments

Editor’s comment

Definition of education. The authors need to clarify how education was defined. The assumption is that the countries included in the study had a similar education system at the time of the surveys, with the same number of years of schooling at each level? If this is not true, then years of schooling may be a more consistent variable than education level.

Authors’ response

Thank you. We agree that education system varies across countries and that is why we did not consider the number of schooling year as the indicator of education. Because length of education in one country does not necessary equivalent to other countries. However, just like wealth index, the categorization of level of education in DHS is a standard measure that allows comparability across surveys (http://dx.doi.org/10.1136/bmjgh-2016-000206). To answers this query and to define education, we have added the following paragraph in the ‘Variable’ subsection (line 145-153 on page 4 of the revised manuscript).

“Education is defined by whether the respondent attended school and if so, the number of years of schooling [51]. Based on this information, DHS provides a standardized variable with four categories mentioned above adjusting for the country-wise education system [52]. The categorization coding is used so that comparison across surveys are possible [53]. Similarly, the wealth index is a standardized measure, quantified using principal component analysis (PCA) from household assets [45].” 

Editor’s comment

What was the extend of missing data (for all the variables included in the analyses)? Could this have affected the observed results? Was the pattern of “missingness” checked? Did you consider using statistical methods to handle missing data?

Please include a STROBE checklist as supplementary file.

Authors’ response

Thank you for the feedback and suggestion. The analysis was conducted after removing case-wise missing values. Since this was a large-scale national survey, we considered listwise deletion with the assumption that data were missing at random for unbiased estimates (Howell, 2007). Furthermore, strata wise sample size was checked and adjusted in the models. We have added the following to address this issue in line 173-175 on page 5 of the revised manuscript.

“As large-scale surveys were used in the study, missing cases were deleted list-wise with the assumption that data were missing at random for unbiased estimates [54].”

We have also added the STROBE checklist as a supplementary file. 

Journal Requirements

http://www.journals.plos.org/plosone/s/file?id=wjVg/PLOSOne_formatting_sample_main_body.pdf and http://www.journals.plos.org/plosone/s/file?id=ba62/PLOSOne_formatting_sample_title_authors_affiliations.pdf.

Authors’ response

We have followed the recommended style.

Authors’ response

We submitted the following information during original submission time of the manuscript.

“This article does not contain any studies with human participants performed by any of the authors. Data used in research was attained from MEASURE Evaluation, funded by the United States Agency for International Development (USAID). All identification of the respondents was dis-identified before publishing the data. Views expressed in this study do not necessarily reflect those of USAID, the US government, or MEASURE Evaluation. The secondary data sets analyzed in the current study are freely available upon request from the DHS website (https://dhsprogram.com/data/available-datasets.cfm). Permission for this project was taken from the Demographic and Health Surveys (DHS) Program authority by the authors.”

We do not have permission to publish the data; however, as indicated above this data is available upon request from the DHS website. We have added the information in the cover letter as well. 

3. We note that Figure 1 in your submission contain map images which may be copyrighted. All PLOS content is published under the Creative Commons Attribution License (CC BY 4.0), which means that the manuscript, images, and Supporting Information files will be freely available online, and any third party is permitted to access, download, copy, distribute, and use these materials in any way, even commercially, with proper attribution. For these reasons, we cannot publish previously copyrighted maps or satellite images created using proprietary data, such as Google software (Google Maps, Street View, and Earth). For more information, see our copyright guidelines: http://journals.plos.org/plosone/s/licenses-and-copyright.

Natural Earth (public domain): http://www.naturalearthdata.com/.

Authors’ response

We have used the ‘maps’ package in R, which is under GPL-2 license. R is an open source software and free to use. Please refer to: https://cran.r-project.org/web/packages/maps/maps.pdf

We have also contacted with the author of the package, Alex Deckmyn (alex.deckmyn@meteo.be). He confirmed that the package used Natural Earth project (www.naturalearthdata.com). This data is in the public domain. They explicitly give permission to use it for all purposes without needing permission (https://www.naturalearthdata.com/about/terms-of-use/). It is also listed above in your suggestions. 

Authors’ response

Thank you for your suggestion. We have followed the suggested style for supporting information and updated this in the revised version of the manuscript.

---

## [Decision Letter · Decision Letter 1]

25 Mar 2020

Women's education and coverage of skilled birth attendance: An assessment of Sustainable Development Goal 3.1 in the South and Southeast Asian Region

PONE-D-19-30198R1

Dear Dr. Biswas,

We are pleased to inform you that your manuscript has been judged scientifically suitable for publication and will be formally accepted for publication once it complies with all outstanding technical requirements.

With kind regards,

Calistus Wilunda, DrPH

Academic Editor

PLOS ONE

Additional Editor Comments (optional):

Reviewers' comments:

Reviewer's Responses to Questions

**Comments to the Author**

1. If the authors have adequately addressed your comments raised in a previous round of review and you feel that this manuscript is now acceptable for publication, you may indicate that here to bypass the “Comments to the Author” section, enter your conflict of interest statement in the “Confidential to Editor” section, and submit your "Accept" recommendation.

Reviewer #2: All comments have been addressed

2. Is the manuscript technically sound, and do the data support the conclusions?

Reviewer #2: Yes

3. Has the statistical analysis been performed appropriately and rigorously? 

Reviewer #2: Yes

4. Have the authors made all data underlying the findings in their manuscript fully available?

Reviewer #2: Yes

5. Is the manuscript presented in an intelligible fashion and written in standard English?

Reviewer #2: Yes

6. Review Comments to the Author

Reviewer #2: Authors have addressed all comments satisfactorily. There is no further comment

7. PLOS authors have the option to publish the peer review history of their article (what does this mean?). If published, this will include your full peer review and any attached files.

Reviewer #2: Yes: Md Zakaria Siddiqui

---

## [Editor Report · Acceptance letter]

27 Mar 2020

PONE-D-19-30198R1 

Women's education and coverage of skilled birth attendance: An assessment of Sustainable Development Goal 3.1 in the South and Southeast Asian Region 

Dear Dr. Biswas:

I am pleased to inform you that your manuscript has been deemed suitable for publication in PLOS ONE. Congratulations! Your manuscript is now with our production department. 

With kind regards,

on behalf of

Dr. Calistus Wilunda 

Academic Editor

PLOS ONE